# An Optimized Workflow for the Analysis of Metabolic Fluxes in Cancer Spheroids Using Seahorse Technology

**DOI:** 10.3390/cells11050866

**Published:** 2022-03-02

**Authors:** Gloria Campioni, Valentina Pasquale, Stefano Busti, Giacomo Ducci, Elena Sacco, Marco Vanoni

**Affiliations:** 1Department of Biotechnology and Biosciences, University of Milano-Bicocca, 20126 Milan, Italy; g.campioni@campus.unimib.it (G.C.); valentina.pasquale@unimib.it (V.P.); stefano.busti1@unimib.it (S.B.); g.ducci@campus.unimib.it (G.D.); elena.sacco@unimib.it (E.S.); 2SYSBIO (Centre of Systems Biology), ISBE (Infrastructure Systems Biology Europe), 20126 Milan, Italy

**Keywords:** cancer metabolism, 3D cultures, high-throughput quantitative live-cell confocal imaging, bioenergetics, mitochondrial respiration

## Abstract

Three-dimensional cancer models, such as spheroids, are increasingly being used to study cancer metabolism because they can better recapitulate the molecular and physiological aspects of the tumor architecture than conventional monolayer cultures. Although Agilent Seahorse XFe96 (Agilent Technologies, Santa Clara, CA, United States) is a valuable technology for studying metabolic alterations occurring in cancer cells, its application to three-dimensional cultures is still poorly optimized. We present a reliable and reproducible workflow for the Seahorse metabolic analysis of three-dimensional cultures. An optimized protocol enables the formation of spheroids highly regular in shape and homogenous in size, reducing variability in metabolic parameters among the experimental replicates, both under basal and drug treatment conditions. High-resolution imaging allows the calculation of the number of viable cells in each spheroid, the normalization of metabolic parameters on a per-cell basis, and grouping of the spheroids as a function of their size. Multivariate statistical tests on metabolic parameters determined by the Mito Stress test on two breast cancer cell lines show that metabolic differences among the studied spheroids are mostly related to the cell line rather than to the size of the spheroid. The optimized workflow allows high-resolution metabolic characterization of three-dimensional cultures, their comparison with monolayer cultures, and may aid in the design and interpretation of (multi)drug protocols.

## 1. Introduction

Metabolic alterations represent one of the hallmarks of cancer [1]. Tumor cells modify their metabolism by enhancing or suppressing conventional anabolic and catabolic pathways as a consequence of tumorigenic mutations (e.g., Ras [2], Myc [3,4]), and/or harsh conditions imposed by the tumor microenvironment [5]. Metabolic rewiring is necessary for cancer cells; they must support an enhanced proliferation rate, requiring a more rapid production of energy and building blocks than normal cells. Moreover, an abnormal vascularization of tumors is often associated with the development of microregions with heterogeneous cells and environments, characterized by high proliferative activity close to the capillaries, quiescent cells as intermediates, and necrotic areas at greater distances from the vessels [6,7].

In vitro cancer models are a fundamental step for studying neoplastic pathology before moving to animal models that are more complex, time-consuming, and cost-intensive—while still necessary [8]. The most utilized in vitro tumor models are bidimensional (2D) monolayers cultures. However, in recent years, it has become clear that these models can behave very differently to cancer in vivo, leading to several failures when moving to animal models [9]. Some of the physiological limits of 2D cultures are, for instance, the bidimensional, monolayer growth on planar surfaces, which reduces cell–cell and cell–extracellular matrix interactions and implies an unrealistic homogeneous distribution of soluble molecules and gases for all the cells in the culture [10]. The development of three-dimensional (3D) cancer models has allowed us to get closer to filling the gap between 2D in vitro models and in vivo animal models because they can better recapitulate the physiological characteristics of cancer. Indeed, the 3D disposition of cells in these cultures causes the spontaneous formation of concentration gradients for oxygen, pH, and soluble components such as nutrients and waste metabolites, leading to a heterogeneous cell phenotype [11]. Among the variety of 3D cancer models, the most reliable to produce and maintain are tumor spheroids. It has been observed that within the spheroid structure, various stages of cells are established due to the gradient described above, including proliferating cells—which can be found mainly at the outer layers of the spheroid—quiescent cells, and even necrotic cells at the central core, due to the hypoxic and starved state to which they are subjected [12,13]. Unlike 2D cultures, the cell heterogeneity represented by 3D models is closer to that observed in cancer in vivo, and it is reasonable to hypothesize that they can be better models for the study of metabolic alterations in cancer.

The employment of Agilent Seahorse technologies for studying metabolism is spreading in many fields, including cancer research [14]. Agilent Seahorse Extracellular Flux (XF) (Agilent Technologies, Santa Clara, CA, United States) analyzers can measure, at the same time, the oxygen consumption rate (OCR) and the extracellular acidification rate (ECAR) of cells in a microplate. It also can monitor variations in these parameters in Real-Time mode after treatment with molecules or drugs of interest, using ad hoc assembled or pre-formulated drug combinations available in kit form. This approach allows the quantitative dissection of the contribution of central metabolic pathways and available nutrients to the fulfillment of the energy requirements of cells. For instance, the Mito stress test includes modulators of the electron transport chain to investigate their role in sustaining cell metabolism. This technology has been exploited, for instance, to perform metabolic phenotyping of cancer cells in standard conditions and under perturbed conditions [15], to discover the metabolic switches responsible for the acquisition of malignant features (e.g., metastasis) [16], to observe metabolic heterogeneity in cancer through the identification of cell subpopulations harboring different metabolic profiles (e.g., cancer stem cells) [17,18], and many other studies of cancer metabolism [19,20,21,22].

However, despite the fact that the most recent and up-to-date XF analyzer, the XFe96 model, is also projected for use in these kinds of studies in three-dimensional cultures, the application of Seahorse XF technology has not been so widely exploited in these models as in bidimensional cultures. Only a few published works have used this technology with cancer spheroids [23,24] and with pancreatic islets from human and mice tissues [25].

This paper proposes an optimized workflow for applying Seahorse XF technology to single cancer spheroids. The workflow involves the generation of a single spheroid in each well of a U-bottom Ultra-Low Attachment (ULA) 96 well plate (PerkinElmer, Waltham, MA, USA). The obtained spheroids are regular in shape (with sections close to a circle) and relatively homogeneous in size, which depends on the number of plated cells. The transfer of these spheroids to XF plates allows the determination of metabolic parameters under basal conditions or following drug perturbations, such as the Mito Stress test. High-resolution imaging allows the calculation of the number of viable cells in each spheroid, the normalization of metabolic parameters on a per-cell basis, and grouping of the spheroids as a function of their size (i.e., the number of viable cells). We tested our complete workflow on two breast cancer cell lines with different metabolic phenotypes: MCF7 and MDA-MB-231 [26,27,28]. Multi-variate statistical tests showed that metabolic differences among the studied spheroids were mostly related to the cell line rather than to the size of the spheroid. The hormone-responsive line MCF7 maintained good metabolic plasticity in 2D and 3D cultures, while the triple-negative MDA-MB-231 withstood metabolic stress much better in 2D than in 3D cultures. The optimized workflow will be helpful in the high-resolution metabolic characterization of 3D cultures, their comparison with monolayer cultures, and for the appropriate design and interpretation of (multi)drug protocols.

## 2. Materials and Methods

### 2.1. Materials and Cell Cultures

The breast cancer cell line MCF7 was a generous gift from Dr. Luca Magnani (Imperial College London). Breast cancer cell lines MDA-MB-231 and SUM159PT were a generous gift from Dr. Lanfrancone (European Institute of Oncology). The bladder cancer cell line RT4 was purchased from American Type Culture Collection (ATCC, Manassas, VA, USA).

The MCF7 cell line was cultured in Dulbecco’s Modified Eagle Medium (DMEM, 11960-044, Gibco™-Thermo Fisher Scientific, Waltham, MA, USA) supplemented with 10% fetal bovine serum (FBS, Gibco™-ThermoFisher, Waltham, MA, USA), 4 mM glutamine, 1 mM Na-Pyruvate, and 10 nM β-estradiol (E2758, Merck Life Science, Darmstadt, Germany), 100 U/mL penicillin, and 100 mg/mL streptomycin.

The MDA-MB-231 cell line was grown in RPMI-1640 medium (R0883-Merck Life Science, Darmstadt, Germany) supplemented with 10% fetal bovine serum (FBS, Gibco™-ThermoFisher, Waltham, MA, USA), 4 mM glutamine, 1 mM Na-pyruvate, 100 U/mL penicillin, and 100 mg/mL streptomycin.

SUM159PT cells were cultured in Ham’s F-12 medium (11765-054, Gibco™-Thermo Fisher Scientific, Waltham, MA, USA) supplemented with 10% fetal bovine serum (FBS, Gibco™-ThermoFisher, Waltham, MA, USA), 2 mM glutamine, 5 μg/mL insulin (I9278, Merck Life Science, Darmstadt, Germany), 1 μg/mL hydrocortisone (H0888-1G, Merck Life Science, Darmstadt, Germany), and 10 mM Hepes, 100 U/mL penicillin, and 100 mg/mL streptomycin.

The RT4 cell line was routinely grown in RPMI-1640 medium (R0883-Merck Life Science, Darmstadt, Germany) supplemented with 10% fetal bovine serum (FBS, Gibco™-ThermoFisher, Waltham, MA, USA), 4 mM glutamine, 100 U/mL penicillin, and 100 mg/mL streptomycin.

All the cell lines were cultured at 37 °C in a humidified atmosphere of 5% CO_2_. Cells were passaged when they reached sub-confluence (typically twice a week) using trypsin-ethylene-diamine-tetraacetic acid (EDTA). The passage number of the cell lines used in the experiments ranged from 10 to 20. The cells were passed for no more than one month (8 passages) after the thawing.

### 2.2. Spheroid Formation Protocols

Spheroid formation was performed in 3D experimental medium DMEM w/o phenol red (Gibco™-Thermo Fisher Scientific, Waltham, MA, USA), 1% BSA, 10 mM glucose, 2 mM glutamine, 10 μg/mL insulin (I9278, Merck Life Science), 0.5 μg/mL hydrocortisone (H0888-1G, Merck Life Science, Darmstadt, Germany), 20 ng/mL EGF (EGF Human Recombinant, PeproTech, London, UK) 100 ng/mL cholera toxin (C8052, Merck Life Science, Darmstadt, Germany), 1 mM Na-pyruvate, 100 U/mL penicillin, and 100 mg/mL streptomycin. Spheroid formation was performed comparing two protocols: the *Multiple Spheroid Protocol* and the *Single Spheroid Protocol*.

*Multiple Spheroids Protocol* (Figure 1a)—the cells were detached using trypsin-EDTA, resuspended in the 3D experimental medium, and seeded in T75 untreated flasks (cod. 658195, Greiner Bio-one Cellstar, Kremsmünster, Austria) at a density of 1.8–2.5 × 10^5^ cells/mL in 10 mL of volume. The flasks were incubated for 3 days (72 h) at 37 °C in a humidified atmosphere of 5% CO_2_ during spheroid formation. The *Multiple Spheroids Protocol* did not include cell staining with CellTracker™ Red CMPTX dye (Invitrogen, Thermo Fisher Scientific, Waltham, MA, USA) because this step was introduced later for simplifying Seahorse normalization.

*Single Spheroid Protocol* (Figure 1a)—spheroids were freshly prepared from the adhesion cultures three days (72 h) before each experiment. The day before seeding (day −1, Figure 1a), the cells grown in a monolayer culture were stained with CellTracker™ Red CMTPX dye (C34552, Gibco^TM^-ThermoFisher, Waltham, MA, USA). First, cells were washed with D-PBS w/Ca and Mg; then, 5 μM CellTracker™ Red CMTPX dye diluted in DMEM w/o phenol red was added and the cells incubated for 30 min at 37 °C and 5% CO_2_. Finally, the dye was removed, cells were washed with D-PBS w/Ca and Mg, and then put in the incubator with their own maintenance medium until the next day. On the day of seeding (day 0, Figure 1a), cells were detached with trypsin-EDTA as usual and centrifuged at 290× *g* for 7 min. Next, the pellet was resuspended in the 3D experimental medium, and cell counting was performed using a Burker chamber. After that, the cells were seeded in U-bottom CellCarrier Spheroid ULA 96-well Microplates (PerkinElmer, Waltham, MA, USA) at a density of 0.5–2 × 10^4^ cells/well in 100 μL/well. After seeding, the U-bottom CellCarrier Spheroid ULA 96-well Microplates were centrifuged at 340× *g* for 30 min to foster cell aggregation. Next, the plate was incubated for 3 days (72 h) at 37 °C in a humidified atmosphere of 5% CO_2_ during spheroid formation.

### 2.3. Imaging Analysis of 3D Models

Image acquisition was performed using the Operetta CLS™ high-content analysis system (PerkinElmer, Waltham, MA, USA) and an Olympus CKX41 (Olympus Scientific Solutions, Tokyo, Japan) phase-contrast microscope when cell culture support differed from multi-well microplates (e.g., petri dish).

Spheroid dimension (area) and roundness were measured using Harmony 4.9 software for images acquired with the Operetta CLS™ system. At least eight stack images with 5× magnification for an overall height of 200 µm were acquired with confocal microscopy, and maximum projection was generated using Harmony software to perform the analyses. Area and roundness were then measured based on the selected fluorescence of the largest section of the spheroid using the ‘calculate morphology proprieties’ analysis block. Morphological analyses of images with 4× magnification acquired using the phase contrast microscope Olympus CKX41, and therefore lacking a fluorescence signal, were performed using ImageJ (https://imagej.nih.gov/ij, U.S. National Institutes of Health, Bethesda, MD, USA), using freehand selection and applying measurements of area and circularity.

### 2.4. Seahorse XFe96 Assay Preparation and Running on 3D Cultures

#### 2.4.1. XFe96 Spheroid Microplate Coating

The day before the assay, all wells of an XFe96 Spheroid Microplate were coated with the adhesive agent collagen type I solution (3867, Merck Life Science, Darmstadt, Germany), diluted in sterile H_2_O at a concentration of 10 µg/cm^2^ (Appendix A Appendix A). A volume of 80 µL of sterile collagen type I solution was dispensed in each well of the XFe96 Spheroid Microplate, avoiding bubble formation. The plate was incubated for 3 h at room temperature; then, the remaining solution was discarded, and the plate was left to dry overnight under the hood. The collagen-coated plate can be stored at 4 °C for up to 7 days. On the day of the assay, the wells were rinsed with sterile H_2_O and left to air-dry before use.

#### 2.4.2. Spheroid Transfer onto the Assay Microplate

Seahorse XF DMEM Medium, pH 7.4 was supplemented with 10 mM D-Glucose and 2 mM L-Glutamine and dispensed 175 µL/well onto the collagen-coated XFe96 Spheroid Microplate.

*Multiple Spheroid Protocol*—the spheroids were harvested in a tube and gently washed once with complete Seahorse XF DMEM Medium. After centrifugation (70× *g*, 5 min), spheroids were resuspended in complete Seahorse XF DMEM Medium. Finally, an aliquot of spheroid suspension was transferred onto a p100 Petri dish. 

*Single Spheroid Protocol*—the medium used for spheroid formation in the U-bottom microplates was half replaced with a complete Seahorse Assay medium to mitigate the buffer presence in the final assay medium.

The Petri dish (*Multiple Spheroids Protocol*) or the U-bottom microplate (*Single Spheroid Protocol*) were placed on a black background to visualize the spheroids better. Single spheroids were picked up using a pre-cut P200 pipette tip or a glass Pasteur pipette to preserve the spheroids’ integrity while retaining only a small amount of media volume (max volume 20 μL). The pipette tip was slowly removed from the pipette and moved against the bottom of an XFe96 Spheroid Microplate well. The spheroid was left to fall upon the center of the well by gravity or by gently tapping the tip. The position of the spheroid was verified by microscope observation and eventually corrected. This procedure was repeated for each well of the XFe96 Spheroid Microplate, excluding background wells. After that, the XFe96 Spheroid Microplate was centrifuged at 340× *g* for 15 min, at low brake. 

Before the assay started, brightfield and confocal fluorescence images of each well were acquired by Operetta CLS to compare the position of each spheroid before and after the assay and to verify if a shift occurred (Appendix A Appendix A).

#### 2.4.3. Sensor Cartridge Hydration and Loading

The day before the Seahorse assay, the sensor cartridge was hydrated with 200 μL/well milliQ water overnight in a non-CO_2_ 37 °C humidified incubator.

On the day of the assay, 1 h before loading the injection ports, the milliQ water was replaced with a pre-warmed Seahorse XF Calibrant solution.

The lyophilized drugs provided in the Seahorse XF Mito Stress Test Kit (Agilent Technologies, Santa Clara, CA, United States) were rehydrated in complete Seahorse XF DMEM Medium to obtain stock solutions: 100 μM Oligomycin, 100 μM FCCP, and 50 μM Rotenone/Antimycin A. The drugs were then diluted in the complete Seahorse medium to reach a 10× solution, to be loaded in the corresponding ports of the sensor cartridge: port A—Oligomycin (20 µL), port B—FCCP (22 µL), port C—Rot/Ant (25 µL). The final concentrations of the drugs were: 2 μM Oligomycin, 2 μM FCCP, and 0.5 μM Rotenone/Antimycin A.

After the Seahorse assay, the medium was carefully discarded with pipette tips from each well, avoiding removal of the spheroids, and the XFe96 Spheroid Microplate was frozen at −80 °C for up to four weeks for protein or DNA content quantification.

#### 2.4.4. Normalization on Area

*Multiple Spheroid Protocol*—immediately after the Seahorse assay, 1 μg/mL Hoechst 33342 (H3570, Gibco-ThermoFisher, Waltham, MA, USA) was added to each well and incubated for 15 min at 37 °C and 5% CO_2_. The fluorescence of nuclei was acquired by Operetta CLS^TM^ (PerkinElmer, Waltham, MA, USA) and analyzed as a single object per well for spheroid area measurements. Harmony software was used for image analysis.

*Single Spheroid Protocol*—the area of each spheroid was determined by selecting the area of cell tracker red fluorescence using the images acquired by Operetta CLS^TM^ (PerkinElmer, Waltham, MA, USA) immediately after the Seahorse assay. Comparison of images acquired before and after the assay allows verification of any spheroid movement from the central position during the Seahorse assay due to the mixing steps before each OCR and ECAR measurement.

#### 2.4.5. Protein Content Assay for Normalization

An XFe96 Spheroid Microplate was thawed at room temperature for protein quantification, and 1N NaOH was added to each well (50 µL/well). The plate was incubated for 20 min at room temperature, and then 50 µL/well of 1N HCl was added. Protein content was determined using a Bio-rad Protein Assay (cat. Number 1-800-424-6723, Bio-rad Laboratories, Hercules, CA, USA); in a 96 well plate, 50 µL of each sample was added to 150 µL of previously diluted 1:5 H_2_O Protein Assay Dye Reagent. Absorbance was read at 595 nm using a FLUOstar^®^ Omega microplate reader (BMG LABTECH, Ortenberg, Germany). The protein content of each sample (including Seahorse Blanks to subtract the contribution of the collagen coating) was derived from a standard curve with Bovine Serum Albumin (BSA). A standard curve with different numbers of cells obtained from the digestion and cell count of a specific number of spheroids was read to calculate the protein content/cell (specifically for each cell line). The number of cells-per-spheroid was determined as the ratio between μg protein/spheroid and μg protein/cell.

#### 2.4.6. DNA Content Assay Normalization

DNA content was quantified using a CyQUANT^®^ Cell Proliferation Assay Kit (Invitrogen-ThermoFisher, Waltham, MA, USA), according to the producers’ manual. A freeze–thaw cycle was added to the protocol to ensure the complete lysis of the spheroids. Green fluorescence was measured using a FLUOstar^®^ Omega microplate reader (BMG LABTECH, Ortenberg, Germany). The DNA content of each sample was derived from a standard curve of the bacteriophage λ DNA standard provided in the kit. A standard curve with cells obtained from the digestion and cell count of a specific number of spheroids was read to calculate the DNA content/cell (specifically for each cell line). The number of cells-per-spheroid was determined by the ratio between ng DNA/spheroid and ng DNA/cell.

### 2.5. Spheroid Digestion and Cell Count

Spheroids were collected in groups of 8–15 spheroids/tube, diluted in D-PBS, centrifuged at 70× *g* for 5 min, and resuspended in 100 μL trypsin-EDTA. Spheroids were incubated at 37 °C for 10 min. Since spheroids tend to precipitate with gravity, after 5 min of incubation with trypsin, the pellet was delicately moved to resuspend the spheroids. Spheroid digestion was facilitated mechanically by pipetting up and down ten times before trypsin neutralization with 200 μL medium with 10% serum. Single cells were counted using the trypan blue exclusion method and a Burker chamber.

### 2.6. Seahorse XFe96 Assay on 2D Cultures

Seahorse assays on 2D cultures were performed according to the manufacturer’s instructions, as reported in Pasquale et al. [15]. MDA-MB-231 and MCF7 cell lines were seeded in Seahorse XF plates at a density of 2 × 10^4^ cells per well and cultured for 24 h in standard culture medium. The next day, the medium was replaced with Seahorse XF DMEM Medium, pH 7.4, supplemented with 10 mM D-Glucose and 2 mM L-Glutamine, and cell cultures were allowed to equilibrate for 1 h at 37 °C in a no-CO_2_ incubator. At the end of the Seahorse measurements, Hoechst 33342 was added to each well at a final working concentration of 1 μg/mL, and after 15 min incubation, nuclei/well were imaged and counted by the Operetta CLS™ software Harmony and directly used to normalize the Seahorse parameters per cell number.

For the Mito stress test, the following drug concentrations were used: 1 μM Oligomycin, 0.25 μM FCCP (MDA-MB-231) or 0.5 μM FCCP (MCF7), and 0.5 μM Rotenone/Antimycin A.

### 2.7. Statistical Analysis

A *p*-value < 0.05 was considered statistically significant. Coefficient of variation calculations, unpaired *t*-tests, and linear regressions were performed using GraphPad version 6 (GraphPad Software, Inc., San Diego, CA, USA).

Multivariate statistical analyses (principal component analysis (PCA) and agglomerative hierarchical cluster analysis) were performed using OriginPro 9.8 (OriginLAb. Corp., Northampton, MA, USA). Data were rescaled into the [0–1] range by min–max normalization as a preliminary step.

PCA was employed to reduce the experimentally observed, possibly correlated parameters to a smaller set of independent variables (principal components, PC), which are linear combinations of original variables that still describe the variance of data, with only a minor loss of information. The cumulative proportion of the variance accounted for by the retained PC1 and PC2 variables was about 87%.

Agglomerative cluster analysis was performed by the ‘average linkage’ method (the distance between two clusters being defined as the average distance between the elements in the clusters). Euclidean distance was employed for measuring the distance between clusters.

## 3. Results

### 3.1. The Single Spheroid Protocol Produces Spheroids Homogeneous in Size and Shape

When using three-dimensional (3D) cultures, the size and heterogeneity of 3D structures, such as spheroids, is the first parameter that can affect the determination of metabolic parameters. This chapter compares two protocols, herein referred to as the *Multiple*
*Spheroids Protocol* and *Single Spheroid Protocol*, respectively.

**Figure 1 cells-11-00866-f001:**
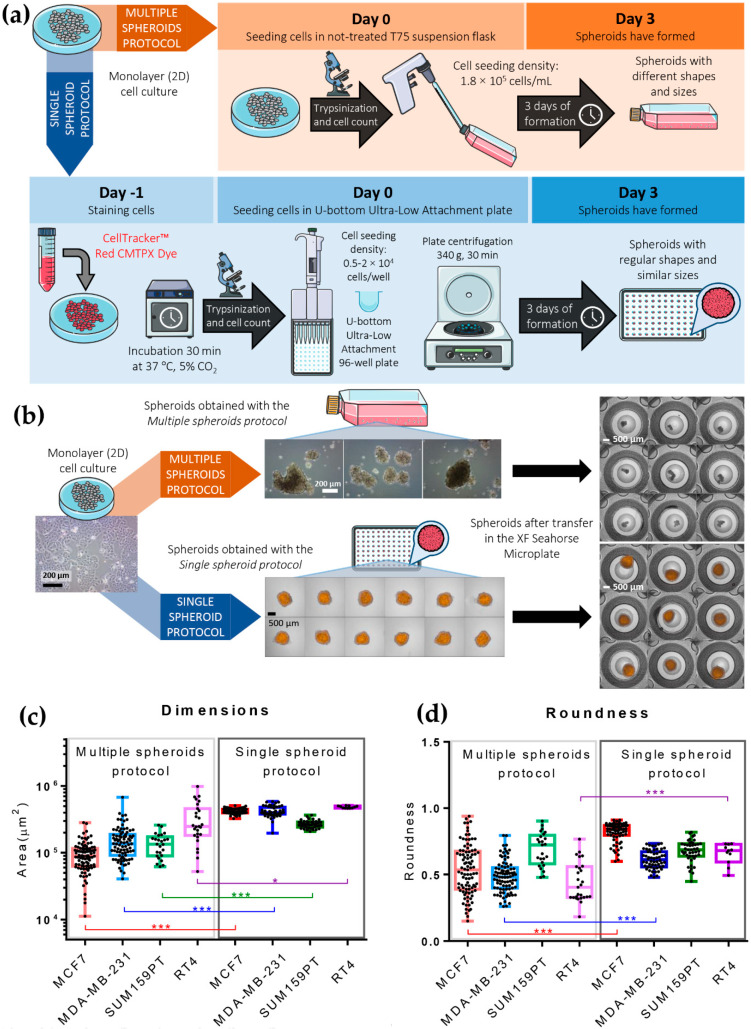
Spheroids from breast and bladder cancer cell lines: protocols for their generation, use in XF Seahorse bioenergetics analyses, and morphometric characterization. (**a**) Schematic representation of the two protocols of spheroid formation: *Multiple Spheroids Protocol* (MSP, above) and *Single Spheroid Protocol* (SPP, below). (**b**) Representative pictures of MCF7-derived spheroids produced with the *Multiple Spheroids Protocol* and the *Single Spheroid Protocol* before and after transfer to the XF Seahorse microplate. Images were acquired with phase-contrast light microscopy (monolayer cell culture and *Multiple Spheroids Protocol* before transfer) and brightfield and confocal fluorescent microscopy (in orange, CellTracker™ Red CMTPX Dye) with Operetta CLS™ (*Single Spheroid Protocol* and spheroids after transfer in the XF Seahorse Microplate). (**c**,**d**) Box and whiskers plots showing single value distribution as black dots for dimension (**c**) and roundness (**d**) of spheroids obtained with *Multiple Spheroids* and *Single Spheroid Protocol* for four different cell lines: MCF7, MDA-MB-231, SUM159PT (mammary carcinoma), and RT4 (urothelial carcinoma); Statistical test: unpaired *t*-test, *** for *p* < 0.001, * for *p* ≤ 0.05.

The *Multiple Spheroids Protocol* exploits the natural tendency of cancer cells to self-aggregate in 3D structures without forcing the formation of regular spheres. It requires the plating of cells suspended in 10 mL of 3D experimental medium in T75 flasks not-treated for tissue culture, at a density of 1.8–2.5 × 10^5^ cells/mL. The spheroids form in 72 h at 37 °C in a humidified incubator, with 5% CO_2_ (Figure 1a). This method—described in more detail in Section 2 is technically straightforward, does not require expensive materials or equipment, and allows the production of many spheroids in a relatively short period (72 h). On the other hand, this method does not allow easy control of the number, dimension, and shape of the formed spheroids.

The *Single Spheroid Protocol* (Figure 1a) consists of seeding the cells (optionally previously stained with CellTracker™ Red CMTPX dye) in U-bottom Ultra-Low Attachment (ULA) 96 well plates in the 3D experimental medium, at a density of 0.5–2 × 10^4^ cells/well in 100 μL/well of medium. After the seeding, the microplates are centrifuged at 340× *g* for 30 min, as described in more detail in Section 2. The plate is incubated for 72 h at 37 °C in a humidified atmosphere of 5% CO_2_ during spheroid formation. The use of ULA U-bottom microplates combined with plate centrifugation after plating fosters cell aggregation in a round shape, causing the formation of a single spheroid per well (Figure 1b). The dimension of the resulting spheroids—formed starting from the same cell number—is relatively homogeneous and can be regulated by varying the number of cells seeded. The *Single Spheroid Protocol* requires more steps and more sophisticated materials than the *Multiple Spheroids Protocol*. 

The spheroids in the untreated T75 flasks (Figure 1b, MCF7 cell line) are morphologically heterogeneous in terms of size and shape, regardless of the cell lines used (Appendix A Appendix A left, MDA-MB-231, SUM159PT, and RT4 cell lines). In contrast, the spheroids obtained in U-bottom CellCarrier Spheroid ULA 96-well Microplates (Figure 1b, below and Appendix A Appendix A right) retain higher regularity. These differences are maintained after their transfer in the Seahorse XF Microplate (Figure 1b, right).

Figure 1c,d and Table 1 report some morphometric parameters determined on the spheroids formed by three breast cancer cell lines (MCF7, MDA-MB-231, and SUM159PT) and the RT4 bladder cancer cell line. The Area defines the measurement of the surface of the largest section of a spheroid. The Roundness defines how closely the shape of the measured object approaches that of a mathematically perfect circle (the section of a sphere); it varies between 0 and 1.

The average Area of spheroids produced with the *Single Spheroid Protocol* (plated at 10,000 cells/well) was between 30% (RT4) and 75% (MCF7) larger than the Area of spheroids produced with the *Multiple Spheroids Protocol* (Table 1, Figure 1c). Even the larger spheroids produced by the *Multiple Spheroid Protocol* were smaller than those produced by the *Single Spheroid Protocol*. The *Single Spheroid Protocol* produced spheroids homogenous in size, with a coefficient of variation (CV) between 8.6 and 17.8, while the CVs of the spheroids obtained with the *Multiple Spheroid Protocol* varied between 39.6 (for the SUM159PT cell line, which is the most prone to form regular spheroids even in untreated plates), and 67.9 (for the bladder cancer RT4 cell line; Table 1). Spheroids produced with the *Single Spheroid Protocol* by all the cancer cell lines analyzed presented with a higher average Roundness and lower intra-spheroid variability than spheroids obtained with the *Multiple Spheroids Protocol* (Figure 1d and Table 1).

These results suggest that the *Single Spheroid Protocol* developed for the formation of 3D cultures produced spheroids more regular (with sections closer to a circle) and homogeneous in terms of size and shape compared to those produced from the application of the *Multiple Spheroids Protocol*. For this reason, the *Single Spheroid Protocol* should provide more reproducible results in the Seahorse assays.

The area of the spheroids produced with the *Single Spheroid Protocol* can be modified without changing the time of spheroid formation through the seeding of different numbers of cells per well at time 0. As reported in Appendix A Appendix A, the area of the spheroids produced with this protocol increased with the number of seeded cells (from 5000 to 20,000 cells/well) and in a cell line-dependent manner.

### 3.2. The Single Spheroid Protocol Allows More Accurate Determination of Oxygen Consumption Rate and Extracellular Acidification Rate by Seahorse XFe96 under Basal and Drug-Perturbed Conditions

Seahorse XFe96 is an extracellular flux analyzer that simultaneously measures the Oxygen Consumption Rate (OCR) and the Extracellular Acidification Rate (ECAR) of cells or spheroids in Seahorse Microplate wells. The OCR is mainly linked to mitochondrial respiration and is only in a small part due to the activity of oxidative enzymes in the cytosol. The ECAR mainly reflects lactic fermentation, but its value is also affected by carbon dioxide released during the TCA cycle, which, combined with water, produces carbonic acid. Therefore, these parameters can provide important information on the metabolic status of the cells in basal conditions and the experimental conditions of interest.

The more regular shape and decreased size variability of spheroids obtained using the *Single Spheroid Protocol* reduced the variability in both the basal Oxygen Consumption Rate (OCR) and basal Extracellular Acidification Rate (ECAR; Figure 2a–d, Table 2).

As shown above, by modifying the number of seeded cells/well, the *Single Spheroid Protocol* modulated the number of cells-per-spheroid, and hence the dimensions of the spheroids produced. The basal OCR (Figure 3a,c) and basal ECAR (Figure 3b,c) of the spheroids were proportional to the number of cells plated to generate the spheroids, indicated in Figure 3 as 5 k (5000 cells/well) and 10 k (10,000 cells/well)—likely reflecting the number of metabolically active cells performing oxidative phosphorylation and fermentative glycolysis during the time of measurement.

The *Single Spheroid Protocol* allowed a reduction in the variability in the size (Figure 1) and metabolic properties of the spheroids (Figure 2). The latter values were generally higher for spheroids formed by plating a higher cell number (Figure 3a–c). As previously mentioned, the dimensional variability of spheroids may impact the access of nutrients and oxygen to the spheroids’ cells, affecting the metabolic properties of the individual spheroids. Direct determination of the actual number of viable cells in each given spheroid using imaging techniques in the XF microplate is not technically practical or even possible, due to the low resolution of the plastic and the high light-scatter of the 3D cultures that prevent a direct count of the cell nuclei inside the spheroids. Moreover, the trypsinization of the single spheroids and manual cell counting using the trypan blue exclusion technique can lead to errors in the estimation of the number of cells-per-spheroid due to the incomplete digestion of the spheroids and the loss of many cells that were alive during the assay but may be dead after the combined treatment with three drugs, as occurs in the Mito stress test.

We reasoned that the spheroid area (a non-invasive measurement that can be obtained by quantitative imaging) could be an appropriate proxy for cell number. Figure 3d shows a high correlation between the spheroid area and the number of viable cells in MCF7-derived spheroids (see Section 2). Using Figure 3d as a calibration curve, it was thus possible to calculate the number of viable cells present in the spheroids, starting from the spheroid’s Area. (Figure 3e). Spheroids formed 72 h after plating contained a higher number of viable cells than those plated at time 0, suggesting that the spheroid formation process is accompanied by some cell division [29]. The number of viable cells present in the spheroids was proportional to the number of cells plated at time 0 in spheroids formed from MCF7 (Figure 3e) and MDA-MB-231 (Appendix A
Appendix A). When basal OCR and ECAR values were normalized on a per-cell basis and clustered according to the actual number of cells present in each spheroid, the difference in basal glycolytic and respiratory fluxes between spheroids of different dimensions faded out, while remaining statistically significant (Figure 3f,g).

### 3.3. The Single Spheroid Protocol Allows More Accurate Determination of Oxygen Consumption Rate and Extracellular Acidification Rate by Seahorse XFe96 Drug-Perturbed Conditions

The Mito Stress Test enables the determination of critical parameters of mitochondrial function by measuring OCR real-time variations during the sequential injection of modulators of oxidative phosphorylation (OXPHOS) into the Seahorse microplate wells. In this assay, three drugs were loaded one after the other in the dedicated ports of the XF sensor cartridge:Oligomycin: causes an OCR decrease due to ATP synthase inhibition. The difference between basal respiration and the lowest OCR value measured after oligomycin injection represents the ATP produced by the mitochondria, contributing to meeting the cell’s energy needs under basal conditions (ATP linked respiration). The difference between the lowest OCR value measured after oligomycin injection and non-mitochondrial respiration (defined below) is called the proton leak and represents the remaining basal respiration not coupled to ATP production. Therefore, it can be a sign of mitochondrial damage or can be used as a mechanism to regulate mitochondrial ATP production.Carbonyl cyanide-4 (trifluoromethoxy) phenylhydrazone (FCCP): disrupts the proton gradient required for ATP synthesis, uncoupling oxygen consumption from oxidative phosphorylation. It increases OCR due to the attempt of the cells to rescue the disrupted mitochondrial membrane potential through the enhancement of electron transport chain activity. This treatment allows the calculation of the Maximal respiration and Spare respiratory capacity, which reflect the capability of the cell to respond to an energetic demand, such as in a stressful condition.A mixture of Rotenone and Antimycin A: these two drugs inhibit complex I and III of the electron transport chain, respectively, enabling the assessment of non-mitochondrial respiration.

Appendix A Appendix A–c report typical Mito Stress test profiles for spheroids formed with the Multiple and *Single Spheroid Protocol* (MCF7 cell line). Not only did the unperturbed, basal OCR of spheroids formed with the *Multiple Spheroids* protocol present with a more considerable variation, as reported above, but different replicates presented with qualitative variations in the Mito Stress profile. On the contrary, spheroids formed with the *Single Spheroid Protocol* presented consistent Mito Stress profiles across the wells. Moreover, as shown in Appendix A Appendix A, the spheroids produced with the *Multiple Spheroids* Protocol were more likely to move during the mixing steps of the assay that occur between one measurement and the subsequent one (this is reported in the Mito Stress profile as a sudden variation of OCR not related to the injection of a drug). This fact may be linked to the smaller dimensions of this group of spheroids, which makes them more sensitive to the mixing process, as they are probably lighter. Therefore, many replicates of the Multiple spheroid protocol had to be excluded from the analysis as they did not remain in a central position in the well throughout the entire assay (Appendix A Appendix A). Accordingly, compared to the Multiple Spheroids Protocol, the *Single Spheroid Protocol* drastically reduced the CV among the replicate spheroids in each of the conditions included in a standard Mito Stress Test (Appendix A Appendix A and Table 3)—the highest variability being observed after the Rotenone/Antimycin A injection.

### 3.4. The Cell Line of Origin Distinguishes the Metabolic Phenotype of Spheroids More Than Their Dimension

Figure 4a shows that the Seahorse Mito Stress test OCR profile of two breast cancer cell lines—MCF7 (left panels) and MDA-MB-231 (right panels)—was strongly influenced by the number of cells seeded to form spheroids (5000–10,000–15,000 cells/well). Normalization to the spheroids’ Area (Figure 4b) or the viable cell number/spheroid (Figure 4c) largely compensated for the differences in basal OCR due to the different dimensions of the spheroids in both cell lines. Notably, when normalized to the viable cell number/spheroid, the Mito Stress profiles of MCF7-derived small and large spheroids were largely superimposable, while MDA-MB-231-derived spheroids showed a more marked dependence of maximal respiration obtained after FCCP treatment on the spheroid’s size (Figure 4c, left and right panels, respectively).

Figure 5a,b present a global overview of the metabolic differences between spheroids formed by the MCF7 and MDA-MB-231 cell lines using principal component analysis (PCA, panels a and b) and agglomerative hierarchical clustering analysis (Panel c). Input data include metabolic parameters determined by the Mito Stress Test (see Appendix A Appendix A) on individual spheroids, normalized to the number of vital cells-per-spheroids determined from the area. First, the number of viable cells present in each spheroid was obtained as described above, and spheroids were grouped according to their cell number in three color-coded classes: (07–10)k class (orange for MCF7 spheroids, cyan for MDA-MB-231 spheroids), (11–15)k class (red for MCF7 and blue (for MDA-MB-231 spheroids) and (16–22)k (black for MDA-MB-231 spheroids). MCF7 spheroids presented with no spheroids in the (16–22)k class. Seahorse parameters were then normalized according to the actual cell number of each spheroid and finally rescaled into the [0–1] range by min–max normalization.

PC1 accounted for 75% of the variability among the metabolic parameters of the spheroids (Figure 5a,b). With only one exception, all MCF7-derived spheroids had positive PC1 values, while the opposite held for MDA-MB-231-derived spheroids. Bigger spheroids tended to have negative PC2 values, while smaller spheroids tended to have positive PC2 values. This behavior was more apparent in MDA-MB-231- than in MCF7-derived spheroids, possibly because this line forms spheroids with a higher number of cells. Nevertheless, since PC2 only accounted for 12.5% of the total variability among spheroids, the difference in size among the spheroids only marginally accounted for the variability in their metabolic properties—at least within the size limits of the spheroids used in these experiments.

Hierarchical clustering (Figure 5c) provided further evidence that metabolic differences among the studied spheroids were mostly related to the cell line rather than to the size of the spheroid. All MCF7-derived spheroids clustered in the left arm, while all MDA-MB-231-derived spheroids (except for one outlier) clustered in the right arm. Accordingly, the values of the measured metabolic parameters were mostly lower for MDA-MB-231 than for MCF7 spheroids, regardless of their respective size (see heatmap in Figure 5c). These data are consistent with the more ‘quiescent’ bioenergetic profile exhibited by the MDA-MB-231 cells (Figure 6 and paragraph below).

### 3.5. Growth in 3D Differentially Affects Metabolic Plasticity in MCF7 and MDA-MB-231 Cancer Cell Lines

Each cell can adjust its metabolic profile according to environmental and/or intracellular demands. Plotting the OCR vs. ECAR values under both basal, unperturbed conditions and conditions of metabolic stress provides a quick overview of the metabolic state of the system under study. Normalizing data from spheroids on a cellular basis as described above allows the direct comparison of metabolic parameters from 2D and 3D cultures. Figure 6 compares the 2D and 3D cultures of MDA-MB-231 and MCF7 cells. Under basal conditions, monolayer cultures of MCF7 (Figure 6a, open triangles) had a more aerobic phenotype than MDA-MB-231 cells [26,28], which showed a more glycolytic metabolism (Figure 6b, open triangles). Growth in 3D drastically altered the metabolic profile of both cell lines. MCF7-derived spheroids (panel a, open circles) showed decreased respiration and increased glycolysis compared to 2D cultures; the effect on respiration was the same regardless of the size of the spheroid, while glycolysis was significantly higher in bigger spheroids. On the contrary, MDA-MB-231-derives spheroids under basal conditions were shifted towards a more quiescent metabolic state, with little if any effect on respiration and a significant reduction in glycolysis (Figure 6b, open circles).

Under stress conditions, obtained by injecting the protonophore FCCP, MCF7 cells showed increased glycolysis and respiration. The 2D cultures (Figure 6a, closed triangles) showed a more significant increase in glycolysis and respiration than the spheroids (Figure 6a, closed circles). The increase in respiratory and glycolytic fluxes following FCCP injection was more pronounced in 2D cultures than in spheroids. The stress condition had little, if any, effect on the metabolic parameters of the MDA-MB-231 2D cultures (Figure 6b, closed triangles). Surprisingly, all MDA-MB-231-derives spheroids showed a significant increase in glycolytic and respiratory flux under stress conditions (Figure 6b, closed circles), suggesting that growth in 3D increases metabolic plasticity in this cell line.

**Figure 6 cells-11-00866-f006:**
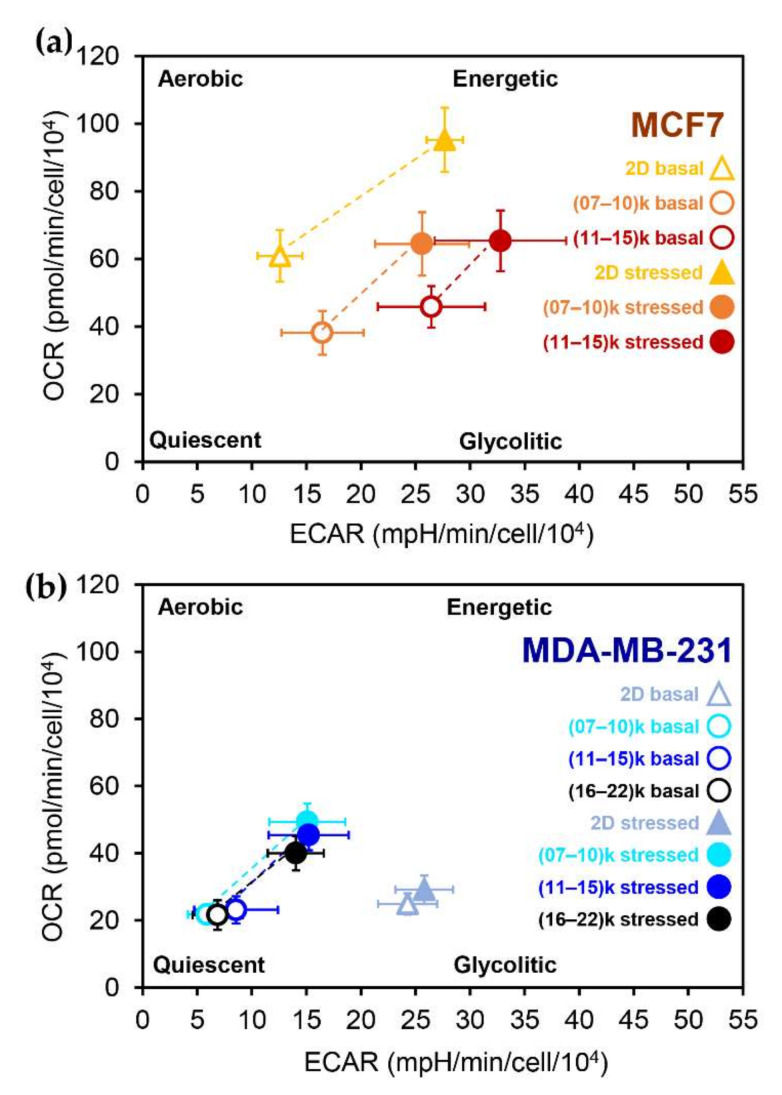
Comparisons of bioenergetic characterization by Seahorse analysis between the 2D and 3D cultures of the MCF7 cell line (**a**) and MDA-MB-231 cell line (**b**), possible due to cell number normalization using imaging techniques, i.e., nuclei counting using Hoechst dye in 2D cultures, area measurement for 3D cultures. Both cell lines were analyzed in three different conditions: growth in monolayer (MCF7 *n* = 37; MDA-MB-231 *n* = 41), spheroids produced with the *Single Spheroid Protocol*, and seeding different numbers of cells per well (5000–10,000–15,000). Spheroids have been re-categorized in 3 groups according to cell number, indirectly calculated through an area-mediated normalization approach: from 7000 to 10,000 cells/spheroid, (07–10) k; from 11,000 to 15,000 cells/spheroid, (11–15) k; from 16,000 to 22,000 cells/spheroid, (16–22) k. Points represent mean ± standard deviation. (**a**) Correlation of OCR and ECAR values under basal conditions and stressed (after FCCP injection) conditions of the MCF7 cell line grown in 2D and 3D; (**b**) Correlation of OCR and ECAR values under basal conditions and stressed (after FCCP injection) conditions of the MDA-MB-231 cell line grown in 2D and 3D.

## 4. Discussion

The use of spheroids and organoids is becoming increasingly popular for studying the properties of tumors and their pharmacological response [11,30,31,32]. In addition, these 3D structures can be valuable pre-clinical avatars in personalized medicine when produced using patient-derived cells [33,34,35,36]. Metabolism integrates information from genetic, epigenetic, and environmental signals so that each physio-pathological condition can be associated with a specific metabolic fingerprinting [37]. Seahorse is one of the most widespread and powerful technologies for dynamically characterizing the metabolism of cells and three-dimensional structures [14,15,22,38,39,40]. Nevertheless, appropriate optimized protocols for preparing spheroids for Seahorse analysis and interpretation of the generated data are still lacking [23,24,25].

This work presents an optimized workflow for producing spheroids of controlled size, regular shape, and reduced inter-spheroid variability that can be easily transferred to Seahorse plates. Since the area of the spheroids—easily determined by quantitative imaging—is proportional to the number of viable cells within the spheroid, it can be used to indirectly determine the number of viable cells that make up the spheroid—thereby obtaining a measure of its size, a highly relevant parameter for properly analyzing Seahorse-derived metabolic data of 3D structures. In fact, in the absence of vascularization, the supply of specific nutrients and oxygen [6] can become limiting for bigger spheroids, up to the point that inner cells may suffer or die [41,42]. Therefore, this normalization approach offers an easy method to correct for differences in viability among different spheroids. Furthermore, since area determination is non-invasive, spheroids can be further processed to generate alternative normalizations. Alternative normalization methods that determine the DNA or protein content produced standard curves that, with one exception, well correlated with the number of plated cells (Appendix A Appendix A). However, compared to the measurement of the Area of the spheroid directly in the Seahorse microplate without further manipulation of the sample, these methods required lysis of the spheroid, pipetting from the Seahorse plate to one suitable for quantification—increasing the experimental error. As a result, they did not provide normalization as accurately as the normalization based on cell number calculated starting from the spheroid Area (Figure 4 and Appendix A Appendix A). Due to the ease of determination and the possibility of studying other parameters of interest, at least qualitatively, we propose using cell numbers calculated starting from the spheroid area to normalize the Seahorse parameters of 3D cultures. Moreover, as discussed later, this normalization approach allows directly comparisons of the metabolic profiles of 2D and 3D cultures.

Pharmacological perturbation of Seahorse-measured parameters provides a multi-faceted view of the metabolic profiles of cell cultures that can be used for high-resolution classification of biological samples. Using two multivariate statistical tests (PCA and agglomerative hierarchical clustering) on parameters calculated from a Mito Stress test performed on individual spheroids of different dimensions produced by MCF7 and MDA-MB-231 cells, we showed that the differences in the metabolic properties of the spheroids were mainly due to the cell line, rather than to their size. This result is not trivial since different papers suggest that the size of the spheroid may affect metabolism qualitatively, inducing rewiring of energy-producing pathways [11,13,43,44]. When the metabolic parameters were not normalized to the number of viable cells comprising each spheroid, spheroids from the two cell lines were not clearly distinguished by either PCA or hierarchical clustering (Appendix A Appendix A). This fact indicates that normalization for the number of viable cells allows the highlighting of differences between the cell lines that the size of the spheroid would otherwise hide.

In the PCA plot, most metabolic parameters (except for ‘spare capacity’ and ‘OCR Rot/Ant’) were positively correlated with PC1, which separated spheroids formed by MCF7 from those formed by MDA-MB-231 cells (Figure 5b and Appendix A). Although PC2 only accounted for less than 13% of the total variability (Figure 5a and Appendix A), it distinguished MDA-MB-231-, and to a lesser extent MCF7-derived, spheroids according to size. The most significant parameter positively affecting PC2 was the spare respiratory capacity (Figure 5b and Appendix A), suggesting that smaller spheroids of both cell lines retained some reserve metabolic capacity to cope with stress conditions. 

MCF7-derived spheroids were more glycolytic (and less respiratory) than 2D cultures (Figure 6a). Bigger spheroids were more glycolytic but did not show a reduction in respiration compared to smaller ones, consistent with the generally accepted view that the compact 3D structure of spheroids limits oxygen availability, forcing cells to increase the glycolytic flux to cope with ATP demands.

The same paradigm did not apply to MDA-MB-231 cultures. When grown in 2D, MDA-MB-231 cells were significantly more glycolytic and less respiratory than MCF7 cells and did not increase metabolism under stress conditions (Figure 6 [26,27,28,45]). Regardless of their size, MDA-MB-231-derived spheroids showed decreased glycolytic flux compared to 2D cultures but—unlike their 2D counterparts—showed the ability to up-regulate metabolism upon stress. So, under basal culture conditions, similarly to MCF7, MDA-MB-231 spheroids showed decreased respiration compared to 2D cultures, in keeping with a possible reduction in oxygen supply to the inner cells of the spheroids—but rather than cope with the reduced respiratory ATP production by increasing glycolysis, they entered a quiescent state, from which they could only partially exit following metabolic stress.

The number of cells per spheroid, its dimension, and its compactness may affect the diffusion gradients of oxygen and other solutes inside the spheroid [46]. In addition, production of spheroids with different protocols (e.g., the addition of matrix that increases the compactness of the spheroid, different times of spheroid formation, etc.) or subject to different treatments (e.g., nutritional deprivations and long-term pharmacological treatments) may significantly alter compactness [11,13,30]). In these cases, appropriate compactness-derived parameter(s) should be considered for normalizing Seahorse parameters and in clustering algorithms.

## 5. Conclusions

Accurate quantitative quality control of spheroids and normalization of data based on the number of viable cells composing the spheroids is mandatory for analyzing the metabolic phenotype of the spheroids. Our results indicate that different cell lines produce spheroids with different metabolic profiles and plasticity. As single-cell technologies [47,48,49,50,51,52,53] progress, it will also be possible to assess the role of cell-to-cell heterogeneity in metabolic properties and drug interactions. Extending the analysis to more cancer cell lines and primary cultures derived from different cancer types (e.g., colon, gastric, lung, skin, ovarian), to heterotypic spheroids including different cell types [54,55]—exploring a broader interval of spheroid dimensions and/or incubation times—may provide an even more complex picture. Finally, 3D structures formed from patient-derived cells [33,34,35] will provide an appropriate platform for designing personalized (multi)drug therapies.

## Figures and Tables

**Figure 2 cells-11-00866-f002:**
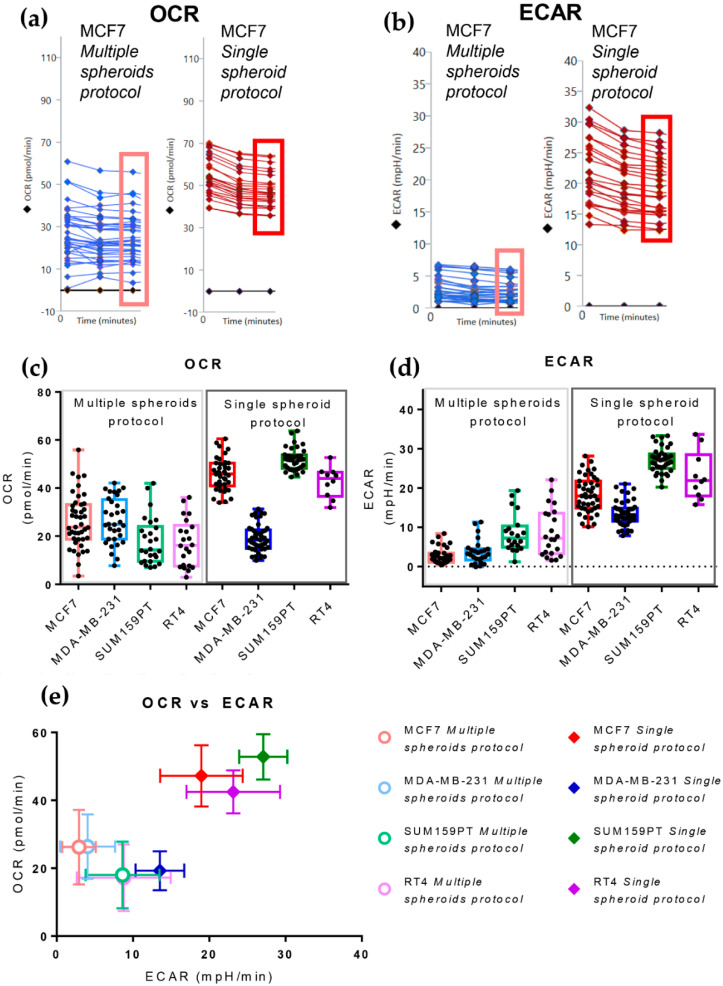
Comparisons of basal Oxygen Consumption Rate (OCR) and ExtraCellular Acidification Rate (ECAR) values between spheroids obtained by the *Multiple* and *Single Spheroid Protocols*. (**a**,**b**) The first three measurements of OCR (**a**) and ECAR (**b**), performed using an XF96 Seahorse Analyzer, showing the baseline profile of MCF7 spheroids obtained with the *Multiple Spheroids Protocol* (left) and *Single Spheroid Protocol* (right). Single measurements for every spheroid are reported to show distribution; the black line is for the background and corresponds to a zero value. The rectangles indicate the 3rd basal OCR and ECAR measurements used to generate the box and whisker plots presented in panels (**c**,**d**). (**c**,**d**) Box and whisker plots comparing basal OCR (**c**) and ECAR (**d**) of four cancer cell lines (mammary carcinoma: MCF7, MDA-MB-231, SUM159PT, and urothelial carcinoma: RT4) produced by applying the *Multiple Spheroids Protocol* and the *Single Spheroid Protocol*; single value distribution is reported as black dots. (**e**) Correlation of OCR and ECAR values under the basal conditions of three mammary carcinoma cell lines produced with the two protocols: MCF7 (MSP *n* = 43; SSP *n* = 45), MDA-MB-231(MSP *n* = 32; SSP *n* = 45), SUM159PT (MSP *n* = 26; SSP *n* = 37), and one urothelial carcinoma cell line: RT4 (MSP *n* = 22; SSP *n* = 10); points represent mean ± standard deviation.

**Figure 3 cells-11-00866-f003:**
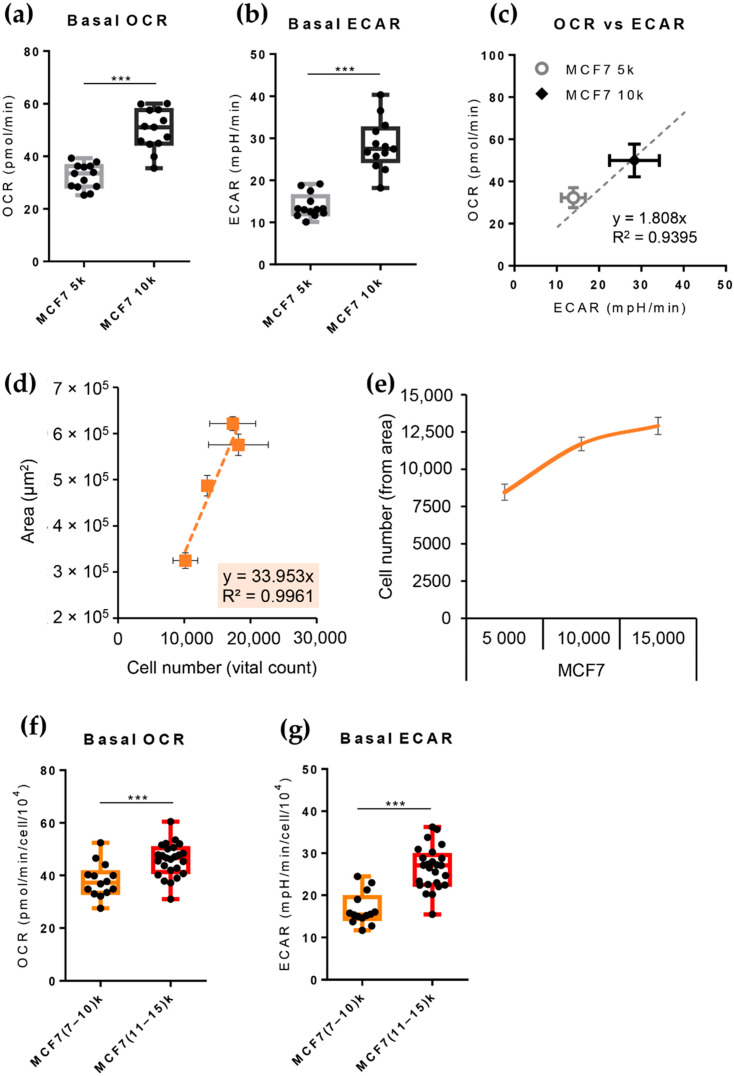
Basal bioenergetic analysis of spheroids obtained with the *Single Spheroid Protocol*, seeding a different number of MCF7 cells per well, and subsequent application of an area-mediated normalization approach. (**a**,**b**) Box and whisker plots of basal OCR (**a**) and basal ECAR (**b**) values of spheroids obtained seeding 5000 (5 k) cells and 10,000 (10 k) cells per well. Statistical test: unpaired *t*-test, *** *p* > 0.001. (**c**) Correlation of OCR and ECAR values (MCF7 5 k *n* = 15, MCF7 10 k *n* = 15); points represent mean ± standard deviation along with regression line. Statistical test: linear regression. (**d**) Standard curve obtained measuring the area of groups of MCF7 spheroids produced from 5000–10,000–15,000–20,000 cells/well and their subsequent digestion and a count of the number of viable cells after 72 h of spheroid formation (note that the number of cells-per-spheroids after 72 h of culture is different than the starting number of cells/well seeded). Points represent mean ± standard deviation along with regression. (**e**) Graph representing the number of cells-per-spheroid calculated indirectly using area measurement on MCF7 spheroids tested in a Seahorse assay, grouped based on the number of cells seeded for spheroid formation (5000–10,000–15,000 cells/well). The points of the orange line depict the number of cells-per-spheroid calculated by measuring the area of each spheroid of the XF Seahorse Microplate and applying the equations derived from the standard curve in panel D. (**f**,**g**) Box and whisker plots of basal OCR (**f**), and basal ECAR (**g**) values of spheroids obtained and re-categorized according to cell number, indirectly calculated through an area-mediated normalization approach. Statistical test: unpaired *t*-test, *** *p* > 0.001.

**Figure 4 cells-11-00866-f004:**
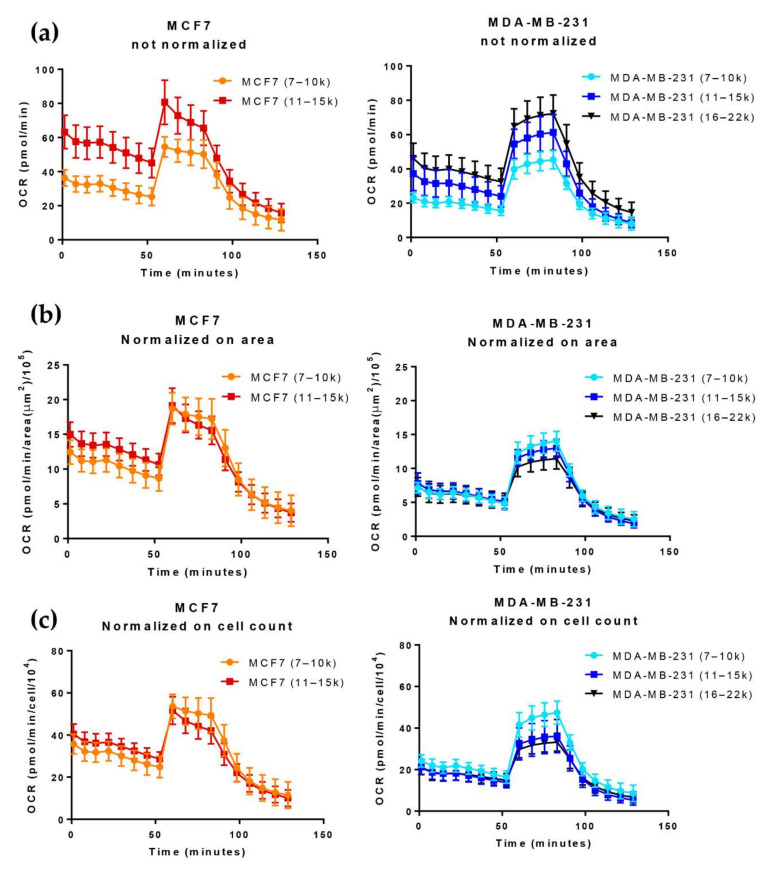
Comparison of the Mito Stress test OCR profile of MCF7 spheroids produced with the *Single Spheroid Protocol,* seeding a different number of MCF7 (5 k and 10 k) cells per well. Spheroids have been re-categorized in 3 groups according to cell number indirectly calculated through an area-mediated normalization approach: from 7000 to 10,000 cells/spheroid, (07–10) k; from 11,000 to 15,000 cells/spheroid, (11–15) k; from 16,000 to 22,000 cells/spheroid, (16–22) k. Points represent mean ± standard deviation. (**a**) non-normalized OCR; (**b**) OCR normalized according to the spheroid area; (**c**) OCR normalized according to the number of cells in the spheroid.

**Figure 5 cells-11-00866-f005:**
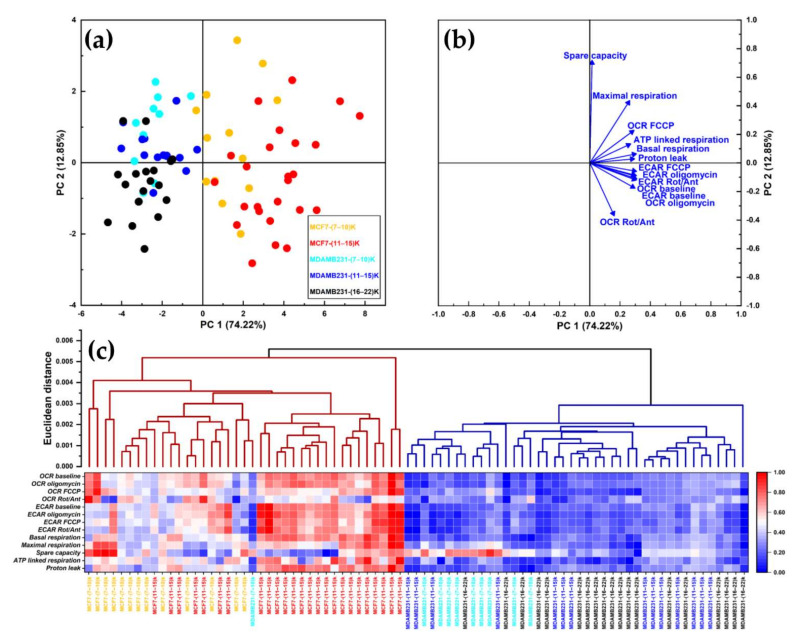
Multivariate statistical analysis for the metabolic parameters of MCF7 and MDA-MB-231 spheroids normalized on the number of cells in each spheroid. (**a**) Principal component analysis (PCA) scores plot. Data are projected onto the two-dimensional space defined by the Principal Component 1 (PC1) and Principal Component 2 (PC2), which together explain about 87% of total data variance. Each symbol represents a single spheroid, color-coded according to the cell line of origin and the cellular counts. (**b**) PCA Loading plot, showing the relationship between the original variables and the Principal Components. (**c**) Hierarchical clustering. The two clusters identified (dark red and dark blue branches) neatly separate spheroids from the two cell lines. The heatmap displays the values of metabolic parameters for each spheroid, scaled to the [0–1] range and color-coded accordingly.

**Table 1 cells-11-00866-t001:** Mean, standard deviation (SD), and Coefficient of Variation (CV) of Area and Roundness, measured on spheroids formed with the *Multiple Spheroids Protocol* and *Single Spheroid Protocol* in four cancer cell lines: MCF7, MDA-MB-231, SUM159PT, and RT4.

Cell Line	Protocol	Area (μm^2^)		Roundness	
		Mean ± SD	CV (%)	Mean ± SD	CV (%)
MCF7	*Multiple spheroids*	95,997.4 (*n* = 93) ± 48,223.0	50.2	0.54 (*n* = 93) ± 0.19	35.2
	*Single spheroid*	421,135.9 (*n* = 75) ± 36,417.5	8.6	0.82 (*n* = 75) ± 0.07	8.7
MDA-MB-231	*Multiple spheroids*	156,971.5 (*n* = 88) ± 94,030.3	59.9	0.49 (*n* = 88) ± 0.12	24.6
	*Single spheroid*	415,612.7 (*n* = 56) ± 73,802.6	17.8	0.61 (*n* = 56) ± 0.07	11.2
SUM159PT	*Multiple spheroids*	137,404.0 (*n* = 25) ± 54,377.4	39.6	0.69 (*n* = 25) ± 0.12	18.1
	*Single spheroid*	267,606.1 (*n* = 45) ± 34,643.6	12.9	0.67 (*n* = 45) ± 0.08	12.2
RT4	*Multiple spheroids*	333,372.0 (*n* = 27) ± 226,341.7	67.9	0.44 (*n* = 27) ± 0.15	33.1
	*Single spheroid*	483,380.8 (*n* = 11) ± 18,691.7	3.9	0.66 (*n* = 11) ± 0.079	12.0

**Table 2 cells-11-00866-t002:** Mean, standard deviation (SD), and the Coefficient of Variation (CV) of Basal Oxygen Consumption Rate (OCR) and Basal Extracellular Acidification Rate (ECAR) measured on spheroids formed with the *Multiple Spheroids* Protocol and *Single Spheroid Protocol* in four cancer cell lines: MCF7, MDA-MB-231, SUM159PT, and RT4.

Cell line	Protocol	Basal OCR		Basal ECAR	
		Mean ± SD	CV (%)	Mean ± SD	CV (%)
MCF7	*Multiple spheroids*	26.20 ± 10.99	41.9	2.68 ± 2.05	76.3
	*Single spheroid*	45.76 ± 6.68	14.6	18.33 ± 4.58	25.0
MDA-MB-231	*Multiple spheroids*	26.36 ± 9.49	36.0	3.72 ± 3.06	82.2
	*Single spheroid*	19.26 ± 5.74	29.8	13.53 ± 3.19	23.6
SUM159PT	*Multiple spheroids*	18.00 ± 9.83	54.6	8.61 ± 4.95	57.5
	*Single spheroid*	52.00 ± 4.69	8.8	27.08 ± 3.16	11.7
RT4	*Multiple spheroids*	17.21 ± 9.82	57.1	8.79 ± 6.16	70.1
	*Single spheroid*	42.48 ± 6.32	14.9	23.14 ± 6.13	26.5

**Table 3 cells-11-00866-t003:** Mean, standard deviation (SD), and the Coefficient of Variation (CV) of Oxygen Consumption Rate (OCR) measured during Mito Stress Test on spheroids formed with the *Multiple Spheroids Protocol* and *Single Spheroid Protocol* (10,000 cells/well) in the MCF7 cell line.

Drug Treatment	Measurement	*Multiple Spheroids Protocol*		*Single Spheroid Protocol*	
		Mean ± SD	CV (%)	Mean ± SD	CV (%)
Basal	1	26.99 ± 12.56	46.5	54.24 ± 9.25	17.1
	2	24.64 ± 11.13	45.2	48.90 ± 8.53	17.4
	3	24.85 ± 11.13	44.8	47.54 ± 8.34	17.5
Oligomycin	4	21.15 ± 9.60	45.4	47.97 ± 8.31	17.3
	5	16.79 ± 8.48	50.5	46.21 ± 8.13	17.6
	6	14.75 ± 7.76	52.6	44.16 ± 7.91	17.9
	7	13.91 ± 7.87	56.6	42.54 ± 7.72	18.2
	8	13.82 ± 8.06	58.3	41.01 ± 7.60	18.5
FCCP	9	39.51 ± 16.53	41.8	73.77 ± 10.58	14.3
	10	35.34 ± 15.04	42.6	75.22 ± 10.41	13.8
	11	32.58 ± 14.29	43.9	76.33 ± 10.35	13.6
	12	30.55 ± 14.48	47.4	77.31 ± 10.28	13.3
Rotenone/Antimycin A	13	17.33 ± 9.01	52.0	60.53 ± 8.86	14.6
	14	12.78 ± 6.92	54.1	38.72 ± 9.19	23.7
	15	11.97 ± 6.89	57.5	28.34 ± 8.50	30.0
	16	11.47 ± 6.35	55.4	22.11 ± 7.55	34.1

## Data Availability

All data are presented in the main text or in Appendix A.

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
