# Peer review of "An Optimized Workflow for the Analysis of Metabolic Fluxes in Cancer Spheroids Using Seahorse Technology"

_cells, 2022, doi:10.3390/cells11050866_

Round 1

Reviewer 1 Report

This is an excellent research article describing an optimized workflow for the analysis of metabolic fluxes in cancer spheroids using seahorse technology. Different experimental conditions offer a variety of implementation aspects potentially highly usable for cancer biology scientists.

Author Response

We thank the reviewer for the appreciation of our work

Reviewer 2 Report

Campioni and colleagues described a workflow for analysis of metabolic fluxes in cancer spheroids using seahorse technology. The authors tested this workflow on two breast cancer cell lines with different metabolic phenotypes, MCF7 and MDAMB231. The authors showed that there are  metabolic differences among the studied spheroids derived from those two lines , and this difference is mostly related to the cell line rather than to the size of the spheroid.  The hormone-responsive line MCF7 maintains 
good metabolic plasticity in 2D and 3D cultures, while the triple-negative MDAMB231 withstands metabolic stress much better in 2D than in 3D.

In general the design is good. However, I have some comments

1- For spheriod formation, I wonder why the authors did not use basement matrix membrane such as matrigel for spheriod formation.

2- Importantly, the passage no could affect on the spheriod formation and hence the results. Please specify the passage no for the cell lines from which the spheriod formed

3-Regarding the spheriod by the two mentioned protocols, how long did the spheriod continue? Can the authors passage these spheriods again? for how many passage? 

4- Importantly, the these spheriods are formed from breast cell lines? Did the authors try the same protocol using colon/ gastric cell lines? If not please mention this in discussion

5- Did the author try this workflow using primary stem cells from breast or other organ??? please specify and this could be a future direction

Reviewer 3 Report

The manuscript entitled ‘An Optimized Workflow for Analysis of Metabolic Fluxes in Cancer Spheroids Using Seahorse Technology’ by Gloria Campioni tried to optimize Seahorse analysis using 3D spheroid culture systems, and the manuscript is well described. Below are minor comments.

1. Line 54: adding ‘heterogeneous’ before ‘cell phenotype’ would be better.

2. For the name of cell line, e.g., MDAMB231, use Hyphens.

3. Cell seeding density 1-4x10^5 was mentioned in the result section. But, in Fig 1, it is differently demonstrated (only 10,000).

4. Check the x-axis of Fig. 3d and S2C again since it was noted that cell number is ranging from 5,000 to 20,000 cells in the Fig legend.

5. Line 537~~: Check the number of Figure panel

6. What is the difference between Fig. 5 and S7?

7. Although authors tried to normalize obtained data using size etc, one of the known characteristics of spheroid 3D culture is the variability of compactness. Therefore, authors need to mention about the possible relationship between spheroid compactness and seahorse analysis in the discussion section.

Round 2

Reviewer 2 Report

The authors replied to my questions adequately.